# Comprehensive Lichenometabolomic Exploration of *Ramalina conduplicans* Vain Using UPLC-Q-ToF-MS/MS: An Identification of Free Radical Scavenging and Anti-Hyperglycemic Constituents

**DOI:** 10.3390/molecules27196720

**Published:** 2022-10-09

**Authors:** Tatapudi Kiran Kumar, Bandi Siva, Ajay Anand, Komati Anusha, Satish Mohabe, Araveeti Madhusudana Reddy, Françoise Le Devehat, Ashok Kumar Tiwari, Joël Boustie, Katragadda Suresh Babu

**Affiliations:** 1Centre for Natural Products & Traditional Knowledge, CSIR-Indian Institute of Chemical Technology, Uppal Road, Tarnaka, Hyderabad 500007, India; 2Academy of Scientific and Innovative Research (AcSIR), Ghaziabad 201002, India; 3Department of Botany, Yogi Vemana University, Vemanapuram, Kadapa 516003, India; 4Faculty of Sciences & IT, Madhyanchal Professional University, Ratibad, Bhopal 462044, India; 5Institut des Sciences Chimiques de Rennes, Université Rennes, CNRS, ISCR-UMR6226, 35000 Rennes, France

**Keywords:** *R. conduplicans*, lichen, secondary metabolites, antioxidant, DNA damage, α-glucosidase inhibition

## Abstract

In this study, we propose ultra-performance liquid chromatography coupled with quadrupole/time-of-flight mass spectrometry (UPLC-QToF-MS/MS)-guided metabolite isolation as a choice analytical approach to the ongoing structure–activity investigations of chemical isolates from the edible lichen, *Ramalina conduplicans* Vain. This strategy led to the isolation and identification of a new depside (**5**) along with 13 known compounds (**1**–**4**, **6**–**14**), most of which being newly described in this lichen species. The structures of the isolates were established by detailed analysis of their spectral data (IR, NMR, and Mass). The acetone extract was further analyzed by UPLC-Q-ToF-MS/MS in a negative ionization mode, which facilitated the identification and confirmation of 18 compounds based on their fragmentation patterns. The antioxidant capacities of the lichen acetone extract (AE) and isolates were measured by tracking DPPH and ABTS free radical scavenging activities. Most isolates displayed marked radical scavenging activities against ABTS while moderate activities were observed against DPPH radical scavenging. Except for atranol (**14**), oxidative DNA damage was limited by all the tested compounds, with a marked protection for the novel isolated compound (**5**), as previously noted for the acetone extract (*p* < 0.001). Furthermore, compound (**4**) and acetone extract (AE) have inhibited intestinal α-glucosidase enzyme significantly (*p* < 0.01). Although some phytochemical studies were already performed on this lichen, this study provided new insights into the isolation and identification of bioactive compounds, illustrating interest in future novel analytical techniques.

## 1. Introduction

Lichens are structurally complex and self-sustaining unique consortia comprised of a fungus host (mycobiont) living with algae or cyanobacteria (photobiont partner) in the framework of a unique symbiotic type of relationship. In recent years, much attention has been paid to the biological roles of lichen metabolites because of their potential applications in perfumery, cosmetics, creative crafts, the dye industry, and the pharmaceutical sector [1,2]. Moreover, many lichens and their extracts and metabolites have been utilized as ingredients in ethnic food preparations and specialties, along with ethnomedicinal applications [3]. For example, a mixture of lichens called Yangben in the Rai and Limbu communities of East Nepal is mainly composed of *Ramalina* species [4]. Among these fruticose epiphytic species, *Ramalina conduplicans* Vain is common and one of the most widely-used edible lichen of the Ramalinaceae family, this is distributed in Central and Southeastern Asian countries [5].

In Southwestern China, people used to prepare their traditional cold dishes with this lichen at their marriage banquets [6], and it also has a long history of consumption as a spice in many places in India and as a traditional food by selected communities in East Nepal [7,8]. In addition to its useful edible properties, crude extracts from this lichen are used as ethnomedicine to counteract inflammation, anthelminthic [9], and act as an anti-diabetic [10], along with antibiotic activities [3,11,12]. Many studies on this lichen have focused on their nutritional value along with the important trace elements [13] and antioxidant properties of *R. conduplicans* [9,14] concerning sekikaic acid and homosekikaic acid [15]. However, systematic investigations of its constituents for their bioactive potentials have not been carried out to date.

Therefore, the antioxidant and alpha-glucosidase inhibiting properties of metabolites from *Ramalina conduplicans* were investigated here as part of our ongoing exploration of natural flora for the isolation of bioactive secondary metabolites [16,17]. Accordingly, we have designed a strategy and workflow based on the Total Ion Current Chromatography (TIC) of the acetone extract (AE) to recognize and to isolate compounds from *R. conduplicans* by UPLC-Q-ToF-MS/MS. AE and all the isolates were assayed for their antioxidant free-radical scavenging properties, including DNA damage protection and anti-hyperglycemic potential, through α-glucosidase inhibition.

## 2. Results and Discussion

The *R. conduplicans* sample was identified by morphological characteristics and thallus reactions: K+ pale yellow, KC−, P−, and also negative reaction of the medulla to calcium hypochlorite solution (C−) (Appendix A). These usual spot tests are based on the presence of lichen metabolites, but have to be supplemented by accurate analytical studies to reveal the metabolite content.

### 2.1. Chemical Profiling, Isolation, and Structure Elucidation

A HPTLC (Appendix A) co-migration with standards and the UPLC-PDA profile (Appendix A) of the acetone extract of *R. conduplicans* suggested the presence of a dozen of visible compounds, among which salazinic acid, usnic acid, sekikaic acid, homosekikaic acid, and divaricatic acid were identified against standards and appeared to represent the most abundant compounds. Initial LC-QToF-MS^E^ analyses of the acetone extract of *R. conduplicans* indicated the presence of depsides, depsidones, and monophenolic acids based on High-Resolution Mass Spectroscopy (HRMS). Molecular formulae for C_10–35_H_10–50_O_2–15_ were generated from mass ranges *m*/*z* 150–750 coupled with the fragment ions and their MS spectral data (accurate mass and fragmentation pattern) and compared to online databases (DNP, Reaxys, SciFinder).

Mass spectrometry (MS) and, particularly, quadruple time-of-flight coupled to Liquid Chromatography (UPLC-Q-ToF-MS) has been widely utilized for profiling metabolites due to its superiority in high-resolution mass, precision, and sensitivity [18], and was helpful to clearly discriminate between the depsides, depsidones, simple phenol acids, dibenzofurans, and hydroxyl fatty acids based on the fragmentation of lichen molecules [16]. Therefore, the acquired TIC of the *R. conduplicans* extracts, obtained within 16 min, were analyzed from spectra obtained in negative mode and, thus, are effective for characterizing trace components (Figure 1). Metabolite assignments were made based on their polarity related to their retention time (Rt) and molecular formulae from accurate molecular weight measurement, along with adducts [M − H]^−^/ fragment ions and Ring Double Bond Equivalence (RDBE). In the present study, a total of 18 compounds were clearly characterized from the crude extract of *R. conduplicans* by molecular formulae generated by ToF-MS/MS and MS/MS including their fragmentation profiles, as reported in the literature and presented in Appendix A.

Based on the fragmentation of isolates, we have identified compounds (**1**–**5**) and (**7**–**9**) along with atranorin belonging to depsides. The literature clearly indicates that sekikaic acid is an abundant molecule in *Ramalina* species [19]. Sekikaic acid (**1**) is a *m*-depside corresponding to the esterification of two divaricatinic acid units and is found with Rt at 11.98 min and *m*/*z* 417.1547 (C_22_H_25_O_8_) with fragments *m*/*z* 209 and *m*/*z* 225 corresponding to the A ring and B ring, respectively [20]. Compounds **1**, **3**, **5**, **7**, **8,** and **9**, having a common fragment *m*/*z* 209 (Appendix A), clearly indicate the difference in locating the other Bring. These depsides can be considered as ester derivatives of divaricatinic acid (**11**) while compound **2** is a divaric acid derivative (recognized at Rt 7.50 min, *m*/*z* 195.0657). The other identified monoaromatic compounds correspond to 2,4-di-*O*-methyldivaric acid (**6**), 4-*O*-methyldivaricatic acid (**10**), divaricatinic acid (**11**), olivetolic acid (**12**), divarinolmonomethylether (**13**), and atranol (**14**). In this run, three additional compounds were ionized and fragmented (Rt = 8.53 min, Rt = 11.88 min, and Rt = 13.17 min) and not determined. The fragmentation feature of Compound **5** (*m*/*z* 401.1954 [M − H]^−^ (calcd. for [C_23_H_28_O_6_]^−^ 401,1964)) suggested the coupling of a divaricatinic acid moiety to an olivetol monomethylether moiety (Appendix A). Based on these fragmentation studies, we assigned compounds as shown in the Supporting Information section and in Appendix A, including the monoaromatic divaric acid, along with the common and already-described atranorin (depside), usnic acid (related to dibenzofurans), and salazinic acid (a depsidone). The structures were concluded through MS/MS fragmentation patterns and compared with in-house standards.

Subsequently, the acetone extract was subjected to column chromatography to give eight fractions (I to VIII). An LC–MS^E^ analysis of all fractions revealed the presence of depsides in III–VI fractions (Appendix A). Thus, the targeted isolation and purification of III–VI fractions yielded the isolation of one new depside (**5**), along with other known depsides (**1**–**4** and**7**–**9**) and monoaromatic compounds (**6** and **10**–**14**). The spectra and fragmentation patterns of these molecules were shown in the Appendix A.

The structures of the isolated compounds (Figure 2) were determined by a combination of spectroscopic data (HRESIMS, ^1^H and ^13^C NMR) and in comparison with the reported literature data. They were identified as sekikaic acid (**1**) [21], 4′-*O*-methylnorhomosekikaic acid (**2**) [22], homosekikaic acid (**3**) [22], hyperhomosekikaic acid (**4**) [23], 2,4-dimethyldivaric acid (**6**) [24], divaricatic acid (**7**) [25], decarboxydivaricatic acid (**8**) [26], decarboxystenosporic acid (**9**) [26], methyldivaricatinate (**10**) [24], divaricatinic acid (**11**) [21], olivetolic acid (**12**) [27], divarinolmonomethylether (**13**) [21], and atranol (**14**) [28].

Compound **5** was isolated as white amorphous powder and identified as a new compound. Its molecular formula was established as C_23_H_29_O_6_ based on a HRESIMS ion at *m*/*z* 401.1954 [M − H]^−^ (calcd. for [C_23_H_28_O_6_]^−^, 401.1964). The ^1^H and ^13^C NMR data of **5** (Table 1) showed the presence of four aromatic protons, (δ_H_ 6.53 (d, *J* = 1.8 Hz, 1H), 6.51 (d, *J* = 1.8 Hz, 1H), 6.46 (d, *J* = 2.6 Hz), and 6.40 (d, *J* = 2.6 Hz); δ_C_ 120.0, 110.7, 105.4, and 100.8) and one ester carbonyl (δ_C_ 170.3). In addition, two methoxyl groups (δ_H_ 3.86 (3H, s), 3.81 (3H, s)) and two n-alkane side chains of two methylene groups that were adjacent to a benzene ring (δ_H_ 3.02–2.93 (m, 2H), 2.62–2.51 (m, 2H)) were also distinguished from the NMR spectra, respectively (Appendix A). These spectral features, together with the characteristic ester carbonyl group at C-7 (δ_C_ 170.3) in the ^13^C NMR spectrum, strongly imply that **5** is a depside-type derivative [16,29].

A comparison of ^1^H NMR and ^13^C NMR data from**5** with those of 4′-*O*-methylnorhomosekikaic acid, which were isolated from the same species, indicated an overall similarity, except for the absence of a COOH group and the presence of two additional methylenes. This reasoning was further supported by its ^13^C NMR spectrum, which showed the absence of a carbonyl COOH group, and its ^1^H NMR spectrum indicated the presence of an additional aromatic proton at 6.53 (d, *J* = 1.8 Hz, 1H). A comprehensive analysis of 2D NMR (COSY, and HSQC) data, especially the ^1^H–^1^H COSY spectrum, revealed two discrete spin systems, including -CH-CH_2_-CH_3_- (from H-1″, H-2″ and H-3″) and -CH-CH_2_-CH_2_-CH_2_-CH_3_ (from H-1′″ to 5′″), as drawn with bold lines in Figure 3. The position of the *n*-pentyl group at C-6′ and *n*-propyl chain at C-6 was confirmed on the basis of the NOESY correlations (H-1′″/H-5′, H-1′″/H-1′ and H-1″/H-5) (Figure 3) and in comparison with the sekikaic acid data. In addition, the MS/MS spectrum of **5** showed (Figure 4) product ions *m*/*z* 209, thereby indicating the breakage of the C–O bond between two aromatic rings supported by the fragments at *m*/*z* 165 and 137. Based on these spectral characteristics, the structure of **5** was established and trivially named as decarboxyhomosekikaic acid.

## 3. Biological Activity

### 3.1. Assessment of Compounds and Extract for Free Radicals Scavenging and Antioxidant Activity

The amphiphilic nature of the ABTS^•+^ cation was used to identify both hydrophilic and hydrophobic antioxidants in dietary materials, whereas the DPPH- radical was used to measure an antioxidant’s reducing power [30]. These fundamental chemical experiments reveal the radical scavenging and reduction characteristics of the potential antioxidant candidates.

Acetone extract (AE) and all isolated compounds (**1**–**14**) scavenged ABTS^•+^ and DPPH^-^ radicals and the results are presented in Table 2. The results have demonstrated that AE and all the other compounds potently neutralized ABTS^•+^ radicals (more than 70%) and have shown activity equal to the ascorbic acid standard, except for compound **13**, which only scavenged radicals by 50%.The pattern and potentials in decreasing order of ABTS^•+^ scavenging potentials were observed as follows:olivetolic acid (**12**) > compound **5** > divarinolmonomethylether (**13**) > decarboxydivaricatic acid (**8**) > 4-*O*-methylnorhomo sekikaic acid (**2**) > atranol (**14**) > divaricatic acid (**7**) > decarboxystenosporic acid (**9**) > sekikaic acid (**1**) > 2,4-dimethyldivaric acid (**6**) > homo sekikaic acid (**3**) > methyldivaricatinate (**10**). In the case of the DPPH-radical scavenging assay, acetone extract (AE) and compound **14** scavenged DPPH-radicals potently by more than 50%, whereas **1**, **2**, **3**, **4**, **5**, **6**, **12**, and **13** countervailed DPPH-radicals (20–40%) moderately. It is important to mention that potent ABTS^•+^ scavenging activities were observed in all compounds, but DPPH scavenging activity was detected to be moderate in all compounds except compound **14** (Table 2). As ABTS^•+^ is a planar radical, it can be used to identify antioxidants even with low redox potentials. However, due to the steric barrier of the N^•^ radical, they may react slowly or not at all when tested on DPPH radicals [31]. This might be the reason why extracts and compounds scavenge ABTS^•+^ more potently than DPPH-radicals. To check whether this radical scavenging activity is related to antioxidant properties, we challenged genomic DNA with hydrogen peroxide (H_2_O_2_)-induced oxidative damage.

### 3.2. Protective Effect of R. conduplicans AE and Isolated Compounds on Oxidative DNA Damage

The Fenton’s reaction produces the hydroxyl radical, which is a ROS that is detrimental to the human body. Hydroxyl radicals react with different nucleobases, thereby inducing the formation of mutated bases that eventually lead to DNA damage [32]. Figure 5 demonstrated that FR damaged DNA significantly (*p* < 0.001) compared to the control (DMSO + DNA). Though all compounds showed significant protection against hydroxyl radical-induced DNA damage (*p* < 0.001, cpd **10**: *p* < 0.05), compound **14** could not prevent the oxidative damage to DNA (Figure 5 and Appendix A). The genoprotective activity of these compounds and the AE may be attributed to the presence of free radical scavenging potential.

### 3.3. Assessment of In Vitro Antihyperglycemic Activity of Compounds and Extract as Intestinal α-Glucosidase Enzyme Inhibition

The α-glucosidase enzyme is a key enzyme that catalyses disaccharide digestion. The inhibition of α-glucosidase in the intestine slows digestion and the overall rate of glucose absorption into the blood. This has proven to be one of the most effective ways for lowering post-prandial blood glucose levels and, as a result, preventing the onset of late diabetes complications [33]. Sekikaic acid (**1**) was already recognized to inhibit α-glucosidase along with usnic acid and salazinic acid from other *Ramalina* species, but it is not the most effective compound [34]. As per Figure 6, it was stated that acetone extract (AE) and compound **4** have displayed better α-glucosidase inhibition (*p* < 0.01) than Acarbose.

On the other hand, compounds **8**, **11**, **12**, and **14** demonstrated inhibitory effects comparable to those of the standard Acarbose (Figure 6). As contrasted activities can be observed between structurally-related compounds, structure–activity relationships can be considered. This is the case between depsides **3** and **4,** suggesting a positive influence of the C1-pentyl chain with regard to substitution by a C1-propyl chain. When this length modification of the alkyl chain occurs on the B ring of decarboxylated derivatives (active compound **8** versus inactive compound **9**) the opposite influence can be observed. The presence of a C6′-carboxylic group lowers the α-glucosidase inhibitory activity as compound **7** is less active than compound **8**. Methylation of the carboxylic function of the monoaromatic divaricatinic acid **11** resulted in a complete loss of activity. Nevertheless, most of the tested compounds were found with some activity, such as compounds **1**, **2**, **3**, **5**, **6**, **7**, and **13,** which displayed mild to moderate enzyme inhibition (*p* < 0.001). These results are to be pooled with the growing number of reports on the antidiabetic potential of lichen extracts or molecules [34,35,36]. The combination of activities with different mechanisms of action is of particular interest to develop potent antihyperglycemic effects. Lowering glucose absorption and limiting oxidative damages due to hyperglycemia, as expected from the lichen extract, could be promising. The challenge is to use standardized extracts that were previously checked to be safe for acute and chronic intake.

## 4. Materials and Methods

### 4.1. General

The NMR spectra were recorded on a Bruker FT-400 MHz NMR spectrometer and samples were dissolved in deuterated acetone-*d_6_*. Mass data were acquired on aXevo^TM^ G2 XS-ESI-QTof mass spectrometer (Waters Corp., Manchester, UK). For thin layer chromatography (TLC) analysis, precoated Merck plates (silica gel 60 F_254_) were utilized. Silica gel (100–200 mesh) (Qing-dao Marine Chemical, Inc., Qingdao, China) was chosen for column chromatographic separation. Semi-preparative chromatography was performed on a Gilson HPLC (Middleton, WI, USA) instrument equipped with a 321 binary pump, GX-281 liquid handler, and UV-155 detector with X Select HSS T3 (250 mm × 100 mm, 5 µm) (Waters Corp., Drinagh, Ireland) as a stationary phase using a Trilution LC v2.1 platform. Formic acid (Optima^TM^ Mass spec grade) (Thermo Fisher Scientific, Geel, Belgium), HPLC-grade acetonitrile, LiChrosolv (Merck, Darmstadt, Germany), and ultra-pure water (Millipore System, Randolph, MA, USA) were used.

### 4.2. Instrumental UPLC Conditions

The instrumental conditions were set-up as per our recent report (Reddy et al., 2019) with slight modifications. Chromatographic separation was performed on an Acquity H Class UPLC system (Waters, Milford, MA, USA) with a conditioned auto sampler using an ACQUITY UPLC CSH Phenyl-Hexyl column (100 mm × 2.1 mm id., 1.7 μm particle size) (Waters, Milford, MA, USA). Column temperature was maintained at 40 °C. High-resolution masses of secondary metabolites were measured after UPLC separation. A mobile phase consisting of water with 0.1% formic acid in water (solvent A) and acetonitrile with 0.1% formic acid (solvent B) was pumped at a flow rate of 0.4 mL/min. The gradient elution program was as follows: 0 min, 5% B; 3.00 min, 20% B; 5.00min, 35% B; 7.50 min, 50% B; 10.00 min, 70% B; 12.50 min, 95% B; 17.00 min 95% B; and 21.00 min 5% B. The equilibration time was 4.0 min and the injection volume was 2 μL. The LC-QTof-MS^E^ mode was applied to analyze the samples in both TIC as well as the MS/MS mode, where the collision energy was ramped at 15–45 eV. Eluted compounds were detected from *m*/*z* 50 to 1200 using a Xevo G2-XS Q-Tof mass spectrometer (Waters, Manchester, UK), which was connected to Electro-spray ionization (ESI) interface with a negative ion mode using the following instrument settings: capillary voltage, 2.0 KV; sample cone, 40 V; source temperature, 120 °C; desolvation temperature 350 °C; cone gas flow rate 50 L/h; desolvation gas (N_2_) flow rate 850 L/h, argon as CID gas for MS/MS experiments. All analyses were performed using lock spray, which ensured accuracy and reproducibility. Leucine–Enkephalin (5 ng/mL) was used as a lock mass, generating a reference ion in the negative mode at *m*/*z* 554.2615, and was introduced by a lock spray at 10 μL/min for accurate mass acquisition. Data acquisition was achieved using MassLynx ver. 4.1. Acquiring data in this manner provided information on intact precursor ions as well as fragment ions.

### 4.3. Lichen Sample Collection and Identification

The lichen, *Ramalina conduplicans,* was collected from tree bark in Bichpuri Range, Bijrani Zone of Corbett National Park, alt. N 29°26′40″ E79°04′06 (1283 m) in the month of May 2019. The morphological features of lichen thallus and ascomata were observed under Magnüs MS 24/13, and spot tests for color reaction were carried out by 10% aqueous solution of potassium hydroxide (K), Steiner’s stable *p*-phenylene diamine solution (PD), and calcium hypochlorite solution (C). For the anatomical investigation of fruiting bodies, a light microscope from ZEISS Axiostar was used. The lichen substances were identified with thin layer chromatography in solvent system ‘A’ following White and James’s methods (1985). The standard literature [37] was referred to for identification of lichen samples. The voucher specimens (Satish Mohabe & A. Madhusudhana Reddy 7658YVUH) of species were deposited at the Herbarium, Department of Botany, Yogi Vemana University, Kadapa, Andhra Pradesh. The corresponding data are shown in Appendix A.

### 4.4. Extraction and Isolation

The sorted-out lichen *Ramalina conduplicans* (300 g) was shade-dried, powdered, and extracted with acetone (6L) at room temperature for 48h. The result was that acetone extract was evaporated to dryness under reduced pressure, thereby affording a syrupy residue (20 g). This crude extract was subjected to gradient column chromatography (SiO_2_, 60–120 mesh) and eluted with a hexane/EtOAc mixture of increasing polarity with 10% intervals that yielded 8 fractions. These eight fractions were reconstituted in acetonitrile and subjected to UPLC Q-ToF MS^E^. Based on the TIC profile, we selected fractions 3–6 for purification (mass profile shown in supporting information, discussion in results section). All these fractions were subjected to semi-preparative HPLC (X Select HSST3 OBD Prep Column,5 μm, 10 mm × 250 mm), 0.1 % formic acid with water (solvent A) and acetonitrile (solvent B) as mobile phase at flow rate 4 mL/min, detected at 254 nm. Semi-preparative HPLC were conducted by gradient elution programs to obtain compounds as follows: Fraction 3 (quantity 70 mg, loop volume 250 μL was eluted by 0 min, 5% B; 5 min, 5% B; 10.00 min, 35% B; 16.00 min, 60% B; 25 min, 95% B; 30 min, 95% B; 5% B; 30.50 min, 5% B; 35.00 min. to yield **3** (homosekikaic acid, 7 mg), **7** (divaricatic acid, 4 mg), **11** (divaricatinic acid, 3 mg), **12** (olivetolic acid, 5 mg), and **14** (atranol, 2 mg). Fraction 4 (quantity 50 mg, loop volume 250 μL was eluted by 0 min, 30% B; 5 min, 30% B; 10.00 min, 50% B; 23.00 min, 95% B; 27.00 min, 95% B; 27.50 min, 30% B; 30% B; 30.00 min at flow rate 4 mL/min, detected at 254 nm to yield **10** (methyldivaricatinate, 3 mg), **8** (decarboxydivaricatic acid, 5 mg), **9** (decarboxystenosporic acid, 2 mg), and **4** (hyperhomosekikaic acid, 1mg). Fraction 5 (quantity 25 mg, loop volume 250 μL was eluted by 0 min, 5% B; 8.50 min, 30% B; 15.00 min, 50% B; 22.00 min, 95% B; 28.00 min, 95% B; 29.0 min, 5% B; 5% B; 35.00 min at flow rate 4 mL/min, detected at 254 nm) to yield **13** (divarinolmonomethylether, 3 mg) and compound **5** (2 mg). Fraction 6 (quantity 40 mg, loop volume 250 μL was eluted by 0 min, 10% B; 8.50 min, 40% B; 18.00 min, 55% B; 25.00 min, 75% B; 32.00 min, 95% B; 36.0 min, 95% B; 10% B; 37.00 min, 10% B; 42.00 min at flow rate 4 mL/min, detected at 254 nm) to yield **1** (sekikaic acid, 5 mg), **2** (4-*O*-methylnorhomosekikaic acid, 7 mg) and 2,4-di-*O*-methyldivaric acid **6** (2 mg). Physicochemical data are shown in the Supporting Materials.

### 4.5. In Vitro Antihyperglycemic and Antioxidant Assay

#### 4.5.1. DPPH Radical Scavenging Activity

A DPPH radical scavenging assay was carried out as previously reported [38]. Scavenging of 2,2-diphenyl-1-picryhydrazyl (DPPH) radicals by the acetone extract (AE) (50 µg of 2 mg/mL solution dissolved in DMSO) and compounds (**1**–**14**) (50 µg of 2 mg/mL solution dissolved in DMSO) was measured in 100 mM Tris-HCl buffer (pH 7.4) by recording the absorbance at 517 nm spectrophotometrically. Ascorbic acid (50 µg of 2 mg/mL solution dissolved in DMSO) served as the standard. The results were expressed as %-scavenging and calculated by using the following formula: (A_c_−A_t_)/100 × A_c_, where A_c_ was the absorbance of control and A_t_ was the absorbance of the test sample. Different concentrations of compounds were evaluated to obtain 50% scavenging activity (SC_50_). The SC_50_ was calculated based on the equation obtained from regression analysis.

#### 4.5.2. ABTS Radical Scavenging Activity

Scavenging of the 2,2′-azino-bis(3-ethylbenzothiazoline-6-sulphonic acid) radical cation (ABTS^•+^) was performed as per the earlier method [39]. Acetone extract (AE) (20 µg of 2 mg/mL solution dissolved in DMSO) and compounds (**1**–**14**) (20 µg of 2 mg/mL solution dissolved in DMSO) were incubated with ABTS^•+^ solution in 6.8 mM phosphate buffer (pH 8.0) as described earlier. The discoloration of the ABTS^•+^ solution was determined by measuring the absorbance at 734 nm spectrophotometrically. Ascorbic acid (20 µg of 2 mg/mL solution dissolved in DMSO) served as the standard. The activity was expressed as %-scavenging and calculated as follows: (A_c_ − A_t_)/(100 × A_c_), where A_c_ was the absorbance of control and A_t_ was the absorbance of the test sample. The SC_50_ of compounds was calculated as per the above formula.

#### 4.5.3. Free Radical Induced DNA Damage

The protective effect of acetone extract (AE) and compounds (**1**–**14**) on oxidative DNA damage was evaluated as per the previous method [40]. A total of 2 µL calf-thymus DNA mixed with 5 µL of 39 mM Tris buffer (pH 7.4) and 5 µL (10 µg) acetone extract and compounds (**1**–**14**) (10 µg of 2 mg/mL solution dissolved in DMSO) mixture was incubated at room temperature for 20 min. The reaction was initiated by adding 5 µL FeCl_3_ (500 µM) and 10 µL H_2_O_2_ (0.8 M) and incubated for 10 min at 37 °C. The reaction was stopped by adding 3 µL DNA loading dye. Finally, the mixture was subjected to 0.8% agarose gel electrophoresis in TAE (40 mM Tris, 20 mM acetic acid and 0.5 M EDTA) buffer (pH 7.2). A total of 3 µL of Ethidium bromide was added to agarose solution to stain DNA bands. The image was viewed under transilluminating UV light and photographed (Bio-Rad, ChemiDocTM XRS, Hercules, CA, USA with Image LabTM software (ver. 6.0.1, build34, standard edition, 2017). The band intensity of the DNA was measured by using ImageJ software (ver. 1.4.3.67, Broken Symmetry Software, Scottsdale, AZ, USA).

#### 4.5.4. Intestinal α-Glucosidase Inhibition

An intestinal α-glucosidase enzyme inhibition assay was performed as per the previous method [36]. A total of 20 μL (40 µg) of acetone extract and compounds (**1**–**14**) (40 µg of 2 mg/mL solution dissolved in DMSO) were incubated with 50 μL of rat intestinal α-glucosidase enzyme (89.93 mM, prepared in 0.9% NaCl) in 100 mM phosphate buffer (pH 6.8) for 10 min. After the incubation period, 50 μL of substrate (4-nitroplenyl α-d-glucopyranoside) solution was added. The release of *p*-nitrophenol from substrate was measured by recording the absorbance at 405 nm spectrophotometrically. Acarbose (40 µg of 2 mg/mL solution dissolved in DMSO) was taken as the standard. The activity was expressed and calculated as follows: (A_c_ − A_t_)/100 × A_c_, where A_c_ was the absorbance of control and A_t_ was the absorbance of the test sample.

#### 4.5.5. Statistical Analysis

Comparisons within the groups were done by applying one-way ANOVA followed by a post-test Tukey’s Multiple comparison test. Statistical significance was set at *p* < 0.05. Data analysis was performed by using GraphPad Prism ver. 5.01 (GraphPad Software Inc., San Diego, CA, USA).

## 5. Conclusions

A novel UPLC-QToF-MS/MS-guided strategy was proposed here for the isolation and characterization of one new depside, decarboxyhomosekikaic acid, along with 13 known metabolites from *Ramalinaconduplicans*—most of them being minor metabolites that were reported on for the first time from this species. In the whole experimental design, UPLC-QToF-MS/MS was selected for multiple purposes, including targeting, finding, profiling, and isolating active constituents. Three hitherto unreferenced compounds were detected in this lichen, with their molecular formulae being deduced from HR-QToF-MS. Although in minute amounts, one isolate could be identified as an additional homosekikaic derivative. The expected major compounds atranorin, usnic acid, salazinic acid, and sekikaic acid were also obtained. However, efforts for isolating, identifying, and testing mainly targeted alkyldepsides- and monoaromatic-related compounds.

These compounds were tested for their antioxidant and α-glucosidase inhibition potential. Most of them, and the crude acetone extract (AE), have displayed antioxidant potential by scavenging ABTS and DPPH radicals and protected DNA from oxidative damage. Five compounds, and particularly hyperhomosekicaic acid, exhibited a comparable or better α-glucosidase inhibition to that of the acarbose standard. On the basis of these results, it is suggested that these lichen substances have a great potential to be used as bioresources or as structural models for novel bioactive candidate compounds. Docking experiments are necessary to document the structure–activities observed in this study along with pharmacomodulation studies to evaluate the antidiabetic properties. Acetone extract unexpectedly showed a comparable effect to that of the Acarbose standard, though it was not sufficient to consider its hypoglycemic activity in the context of the traditional use made of this edible lichen [10].

It should be kept in mind that activities obtained from the crude extract or from any of the active metabolites cannot be claimed to support a preventive or a therapeutic activity as no clinical assay has been carried out to validate an effect with a standardized dosage. Unexpected side effects can occur when preparations differ from the real traditional use, and toxicity trials have to be carried out at once.

## Figures and Tables

**Figure 1 molecules-27-06720-f001:**
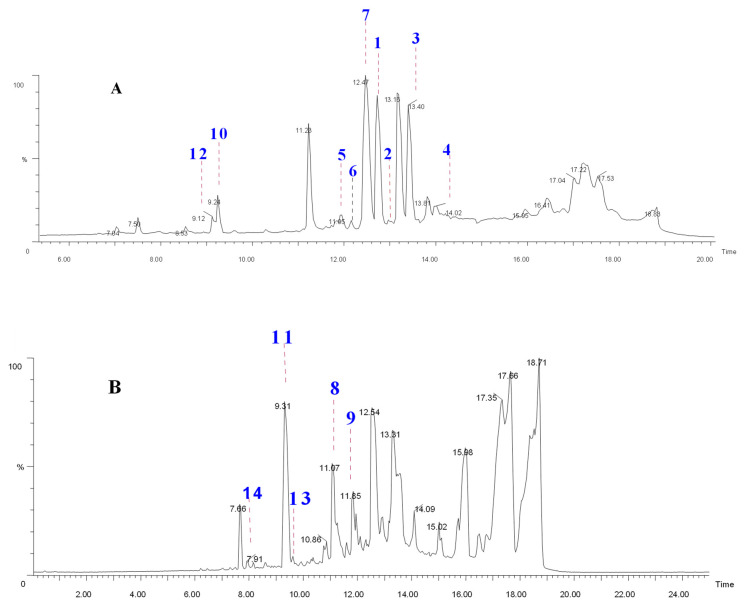
TIC of (**A**) *R. conduplicans* acetone extract and (**B**) enriched fraction-4.

**Figure 2 molecules-27-06720-f002:**
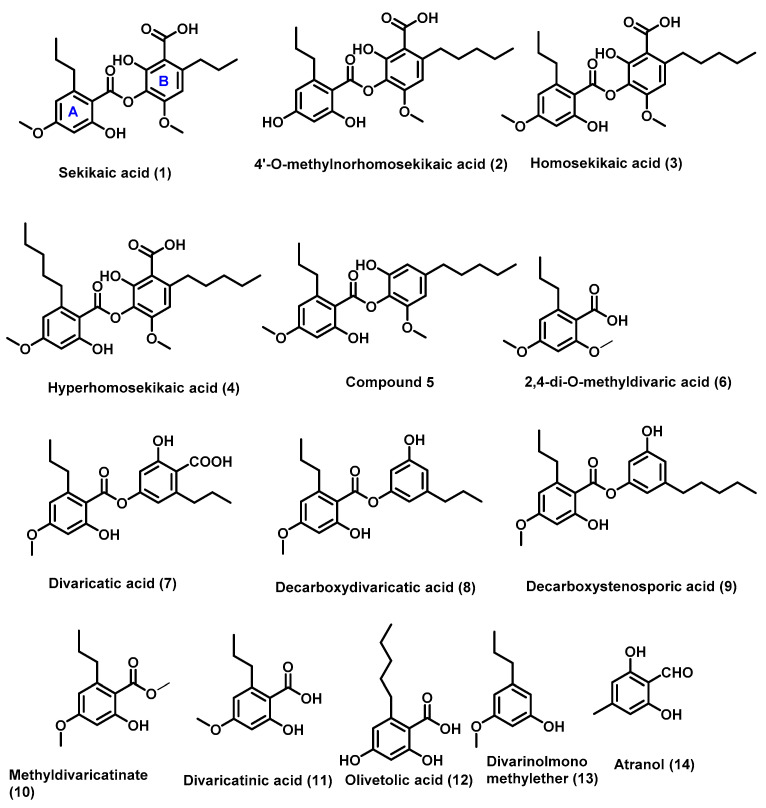
Isolated compounds (**1**–**14**) from *Ramalina conduplicans* Vain.

**Figure 3 molecules-27-06720-f003:**
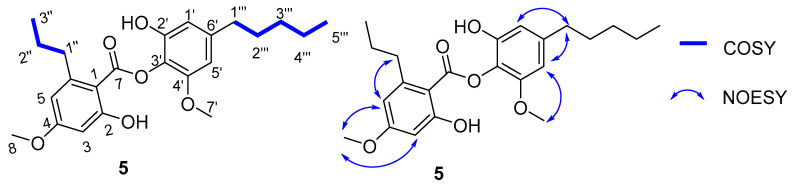
Key COSY and NOESY correlations of compound **5**.

**Figure 4 molecules-27-06720-f004:**
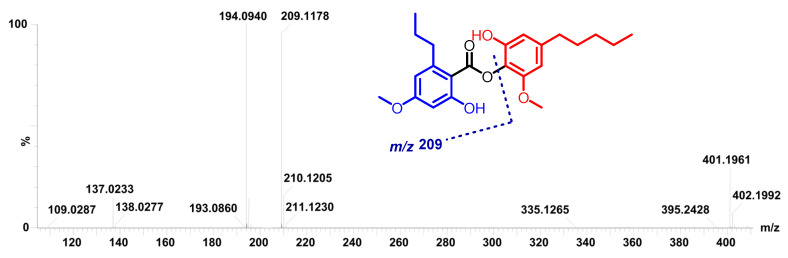
MS/MS spectrum and proposed fragmentation of compound **5**.

**Figure 5 molecules-27-06720-f005:**
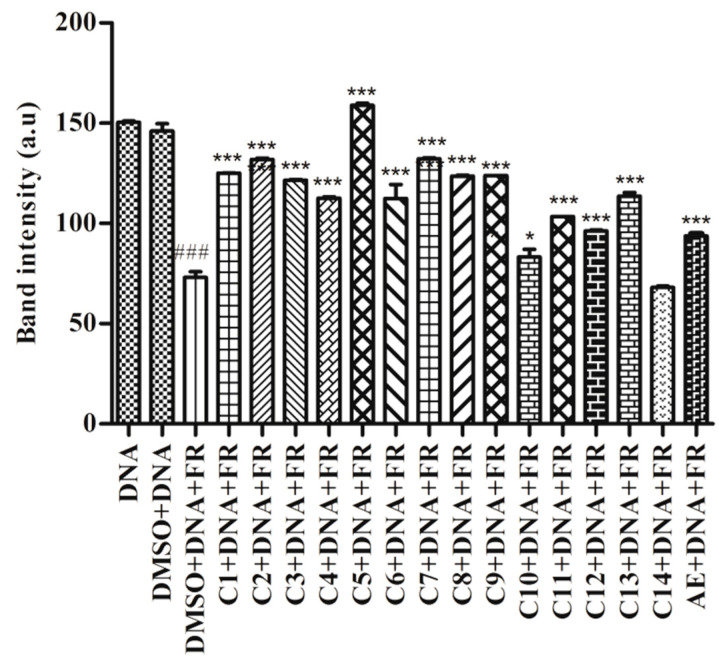
In vitro DNA damage assay. Compounds (**1**–**14**) and *R. conduplicans.* AE were incubated with DNA and Fenton’s Reagent and DNA damage was recorded with Agarose Gel electrophoresis. Respective graphical representation. ### *p* < 0.001; vs. control (DMSO + DNA). *** *p* < 0.001, * *p* < 0.05; vs.DMSO + DNA + FR, One-way ANOVA followed by Tukey’s multiple comparison test was used to calculate values. Values are represented as mean ± SD, *n* = 3. AE = Acetone Extract, FR = Fenton’s Reagent.

**Figure 6 molecules-27-06720-f006:**
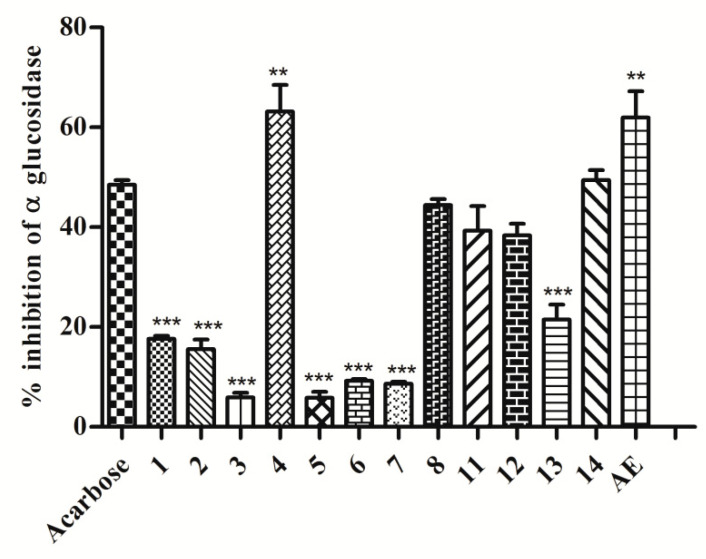
Intestinal α-glucosidase inhibitory assay. Compounds (**1**–**8**), (**11**–**14**), and acetone extract (AE) were incubated with α-glucosidase enzyme and the release of *p*-nitrophenol was determined. *** *p* < 0.001, ** *p* < 0.01; vs. Acarbose. One-way ANOVA followed by Tukey’s multiple comparison test was applied to compare differences. Values are represented as mean ± SD, *n* = 3. AE = Acetone Extract. Activity was not detected for compounds **9** and **10**.

**Table 1 molecules-27-06720-t001:** NMR data of compound **5** (400 & 100 MHz, acetone-*d_6_*) *.

S no	^1^H NMR of 5	^13^C NMR of 5
1	--	107.35
2	--	166.10
3	6.46 (d, *J* = 2.6 Hz, 1H)	112.07
4	--	166.10
5	6.40 (d, *J* = 2.6 Hz, 1H)	100.65
6	--	149.91
7	--	170.47
1′	6.53 (d, *J* = 1.8 Hz, 1H)	110.80
2′	--	151.36
3′	--	154.12
4′	--	143.54
5′	6.51 (d, *J* = 1.8 Hz, 1H)	105.49
6′	--	150.38
1″	3.0–2.93 (m, 2H)	40.06
2″	1.82–1.68 (m, 2H)	26.58
3″	0.93 (t, *J* = 7.6, 3H)	15.60
1‴	2.62–2.51 (m, 2H)	37.69
2‴	1.67–1.59 (m, 2H)	32.89
3‴	1.41–1.30 (m, 2H)	33.25
4‴	1.41–1.30 (m, 2H)	24.18
5‴	0.93 (t, *J* = 7.6 Hz, 3H)	15.30
OMe-7′	3.81 (s, 3H)	57.31
OMe-8	3.86 (s, 3H)	56.83

* = values are assigned with the comparison of sekikaic acid data and COSY/NOESY correlations.

**Table 2 molecules-27-06720-t002:** Free radical scavenging activities of AE and compounds (**1**–**14**) of *Ramalina conduplicans*.

Compound Name (Code)	DPPH Assay % Scavenging (SC_50_, µg/mL)	ABTS Assay % Scavenging, (SC_50_, µg/mL)
Sekikaic acid (**1**)	37.75 ± 0.65	99.05 ± 0.00 (2.45)
4-*O*-methylnorhomosekikaic acid (**2**)	36.78 ± 1.57	98.57 ± 0.00 (1.40)
Homosekikaic acid (**3**)	38.28 ± 1.22	98.10 ± 0.00 (2.81)
Hyperhomosekikaic acid (**4**)	27.32 ± 0.34	79.52 ± 1.02 (17.44)
Compound **5**	46.29 ± 3.70	100.48 ± 0.68 (0.44)
2,4-dimethyldivaric acid (**6**)	29.67 ± 0.89	100.20 ± 0.34 (2.46)
Divaricatic acid (**7**)	8.57 ± 1.65	99.28 ± 1.02 (2.09)
Decarboxydivaricatic acid (**8**)	17.17 ± 1.74	100.28 ± 0.00 (0.75)
Decarboxystenosporic acid (**9**)	13.04 ± 0.73	96.7 ± 0.5 (2.41)
Methyl divaricatinate (**10**)	6.50 ± 0.76	100.0 ± 1.0 (2.90)
Divaricatinic acid (**11**)	ND	100.0 ± 0.5 (2.63)
Olivetolic acid (**12**)	41.94 ± 1.11	92.7 ± 0.0 (0.13)
Divarinolmonomethylether (**13**)	21.12 ± 1.21	51.7 ± 2.5 (0.57)
Atranol (**14**)	74.66 ± 2.59 (18.65)	88.9 ± 7.3 (2.05)
Acetone Extract (AE)	50.64	97.3
Ascorbic Acid	93.25 ± 1.23 (3.96)	99.02 ± 0.03 (0.47)

ND = Not determined. The activity is expressed as % scavenging with regard to ascorbic acid scavenging activity. The SC_50_ is indicated for the most active compounds.

## Data Availability

The data presented in this study are available on request from the corresponding authors.

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
