# Peer review of "Comprehensive Lichenometabolomic Exploration of Ramalina conduplicans Vain Using UPLC-Q-ToF-MS/MS: An Identification of Free Radical Scavenging and Anti-Hyperglycemic Constituents"

_molecules, 2022, doi:10.3390/molecules27196720_

Round 1

Reviewer 1 Report

The paper submitted to Molecules is focused on ultra-performance liquid chromatography coupled with quadrupole/time-of-flight mass spectrometry (UPLC-QTOF-MS/MS) guided isolation of secondary metabolites from the edible lichen, Ramalina conduplicans. The subject of the research seems interesting and worth investigation, however the paper needs substantial improvement before it can be considered for publication in Molecules. Below please find my comments:

1) Remove all unnecessary designations from chromatograms or redraw them in a graphical editor

2) In table 1, correct the positions of aromatic protons. In the text on lines 151-153 “13C NMR spectrum which showed absence of carbonyl COOH group and 1H NMR spectrum indicated the presence of an additional aromatic proton at 6.46 (1H, d, J = 2.6 Hz)” in table 1, the proton with the corresponding chemical shift is designated as 3 although it should be 1’ (Fig 3). The same table shows a proton at 3’ although there are no protons

3) How the SC50 value was calculated? There is no information about this in the article.

4) Statistical analysis should be added to the Materials and Methods section

5) Figure 5A (Agarose Gel electrophoresis) can be removed to the Supplementary Materials, as it is less informative than Figure 5B

6) It is necessary to consider the antiradical activity of isolated substances from the point of view of structure-activity, as is done in the case of the α-glucosidase enzyme

7) Line 247 it is necessary to correct the column diameter from 100 mm to 10 mm

8) The voucher number of the studied lichen specimens in the herbarium is not provided

Reviewer 2 Report

The paper presents a detailed biochemical and analytical analysis of Ramalina conduplicans. Extensive studies have also been carried out to show the identification of a novel phytochemical as well as its biological activity. However, I have some major concerns about the study.

Firstly, the authors mention that the samples were collected in November 2017. Were these studies conducted in 2017? If not, how were the samples stored?, how stable are the phytoextracts? Ideally, phytochemicals are stable for 6-12 months if the samples are freeze-dried.

Following, acetone extraction, the samples were dried to evaporation. But no mention about what solvents were the samples reconstituted in is not mentioned. For the LCMS analysis of the column chromatography extracts eluted with Hexane/EtOAc, directly injected into the LCMS? The choice of solvent for sample reconstitution has significant bearing on the overall success of natural product isolation!

For LCMS analysis, what was the rationale for going with 0.1% FA in both Solvent A and B when the analysis was to be carried out in negative mode. Presence of FA is known to cause ion suppression and higher propensity of FA adduct formation in negative ionization mode. Despite this, the authors only examined [M-H]- adducts in the m/z spectrum. 

For NMR experiments, no NMR operating parameter are reported. Also, for HPLC-UV experiments, since the authors did not mention what the samples matrix was, I have questions on how did the authors take into account background UV spectra? Organic solvents and other plant metabolites also can absorb UV in the range of 220-260nm     

Reviewer 3 Report

Dear Authors,

The manuscript Comprehensive Lichenometabolomic Exploration of Ramalina conduplicans Vain using UPLC-MS/MS: Identification of Free Radical Scavenging and Anti-Hyperglycemic Constituents that you submitted to the journal Molecules is very interesting and original. In my opinion, this manuscript would be of interest not only to readers of the journal Molecules but also to other readers.

In my opinion, the Introduction part has given you complete information about research by them food lichen Ramalina conduplicans. Its benefits to human health as a food product and antioxidant.

 In the Methods and Materials section, I think it is written very precisely and any researcher could repeat the experiments done by the authors. I would also like to add that it is desirable that other researchers repeat the experiment in order to really prove the new compound that the authors obtained. I would also like to ask the authors to be consistent in writing the abbreviation of the device they use. I'll give you an example of what I mean UPLC-Q-TOF-MS/MS, UPLC-Q-Tof-MS/MS and other variants. I think that in the title of the manuscript they should correct the abbreviation of the apparatus.

And last but not least is the Results and Discussion section. In my opinion, the discussion made by the authors is completely accurate and supports the results obtained. The authors successfully prove that their new compound 5 was isolated and proved by means of NMR, IR, Mass Spectoscopy. Based on the authors' spectral characteristics of this compound, they gave the following name decarboxyhomoseic acid.

In this section Results and discussion, I would like to make a suggestion to the authors, let the figures that are black and white be colored for clarity, so that the readers themselves will immediately notice the differences.

In my opinion, the conclusion fully supports the results obtained by the authors.

Reviewer 4 Report

The manuscript (Comprehensive Lichenometabolomic Exploration of Ramalina  conduplicans Vain using UPLC-MS/MS: Identification of Free  Radical Scavenging and Anti-Hyperglycemic Constituents) idea is good but for more improving the quality, I suggest some suggestions

- I didnt saw any quantitative analysis for the major compunds or the identified acids or compounds why 

- The results of Table 2 expressed by % Scavenging (SC50, μg/mL) is it % or W/L

-The discussion of table 2 needs more explanations and references updated

- The reference used needs to updated for last two years

- The date of Ramalinaconduplicansvwas collected since 2017, I dont know if the work done after extraction the compounds or in this year. I afraid of the analysis done from old samples or why the data didnt published from 2017

- The conclusion must re-write by the most important results in the manuscript, The most important beneficiaries of these results and the most important applications that will use these results and the future vision to benefit from this research and reference 39 is removed and used in the discussion

Round 2

Reviewer 2 Report

I understand the idea of not going to extensive details in the materials/methods in the main text, but in the interest of science and reproducibility, it is necessary to have it clearly listed or following a previously detailed procedure, then it needs to be cited. In the present case, it would be good to have the experimental methods clearly described in the Supporting Information

Author Response

I understand the idea of not going to extensive details in the materials/methods in the main text, but in the interest of science and reproducibility, it is necessary to have it clearly listed or following a previously detailed procedure, then it needs to be cited. In the present case, it would be good to have the experimental methods clearly described in the Supporting Information

  •  As suggested by the reviewer, we have reported Materials and Methods section of MS with additional details to the first part of Supporting Information. These additions are in red color to be visible and concern NMR details and UV absorbance details for the HPLC separation process. Related references are at the end of the SI.

  • In the MS, the references were re-checked, reformatted and corrected wherever required.

We removed reference 39 which was not found anymore relevant 

39. Jha, B. N.; Shrestha, M.; Pandey, D. P.; Bhattarai, T.; Bhattarai, H. D.; Paudel, B.; Investigation of antioxidant, antimicrobial and toxicity activities of lichens from    high altitude regions of Nepal. BMC Complement Altern. Med. 2017, 17, 1-8.

and added the following ones to support the discussion on the DNA damage assay (ref 32) and the identification of the lichen (ref 37.)

32. Lloyd, D. R., Phillips, D. H. Oxidative DNA damage mediated by copper (II), iron (II) and nickel (II) Fenton reactions: evidence for site-specific mechanisms in the formation of double-strand breaks, 8-hydroxydeoxyguanosine and putative intrastrand cross-links. Mutation Research/Fundamental and Molecular Mechanisms of Mutagenesis, 1999, 424(1), 23-36.

37. Awasthi, D.D. A Compendium of the Macrolichens from India, Nepal and Sri Lanka. DehraDun: Bishen Singh Mahendra Pal Singh. 2000.  

So Refs 32 to 40 were adjusted accordingly in the MS

  • We revised in our best many parts to improve style and for English spelling. Many are from the suggestions of a native English speaker (Dr S. Labarre, mentioning his contribution in Acknowledgements)

Reviewer 4 Report

I suggest to publish this manuscript in present form

Author Response

Thanks for your time, your remarks and support to improve the MS

We revised in our best many parts to improve style and for English spelling. Many are from the suggestions of a native English speaker (Dr S. Labarre, mentioning his contribution in Acknowledgements)